# The protective effect of carbamazepine on acute lung injury induced by hemorrhagic shock and resuscitation in rats

Yaqiang Li[1], Hiroko Shimizu[2☯*], Ryu Nakamura[2], Yifu Lu[1¤], Risa Sakamoto[1], Emiko Omori[1], Toru Takahashi[3], Hiroshi Morimatsu[1☯]

1 Department of Anesthesiology and Resuscitology, Graduate School of Medicine, Dentistry and Pharmaceutical Sciences, Okayama University, Okayama, Japan, 2 Department of Anesthesiology and Resuscitology, Okayama University Medical School, Okayama, Japan, 3 Okayama Saidaiji Hospital, Okayama, Japan

☯ These authors contributed equally to this work.
¤ Current address: Department of Human Anatomy, Shantou University Medical College, Shantou City, Guangdong Province, China
* hirokos@md.okayama-u.ac.jp

**Data Availability Statement:** All relevant data are within the manuscript and its Supporting Information files.

## Abstract

Hemorrhagic shock and resuscitation (HSR) enhances the risk of acute lung injury (ALI). This study investigated the protective effect of carbamazepine (CBZ) on HSR-induced ALI in rats. Male Sprague-Dawley rats were allocated into five distinct groups through randomization: control (SHAM), saline + HSR (HSR), CBZ + HSR (CBZ/HSR), dimethyl sulfoxide (DMSO) + HSR (DMSO/HSR), and CBZ + chloroquine (CQ) + HSR (CBZ/CQ/HSR). Subsequently, HSR models were established. To detect tissue damage, we measured lung histological changes, lung injury scores, and wet/dry weight ratios. We measured neutrophil counts as well as assessed the expression of inflammatory factors using RT-PCR to determine the inflammatory response. We detected autophagy-related proteins LC3II/LC3I, P62, Beclin-1, and Atg12-Atg5 using western blotting. Pretreatment with CBZ improved histopathological changes in the lungs and reduced lung injury scores. The CBZ pretreatment group exhibited significantly reduced lung wet/dry weight ratio, neutrophil aggregation and number, and inflammation factor (*TNF-α* and *iNOS*) expression. CBZ changed the expression levels of autophagy-related proteins (LC3II/LC3I, beclin-1, Atg12-Atg5, and P62), suggesting autophagy activation. However, after injecting CQ, an autophagy inhibitor, the beneficial effects of CBZ were reversed. Taken together, CBZ pretreatment improved HSR-induced ALI by suppressing inflammation, at least in part, through activating autophagy. Thus, our study offers a novel perspective for treating HSR-induced ALI.

## Introduction

Acute lung injury (ALI) is a significant cause of mortality in critically ill individuals, accounting for over 10% of all intensive care unit admissions [1]. One of the most common precipitating factors of ALI is hemorrhagic shock and resuscitation (HSR) [2–4]. Our previous studies

**Funding:** This work was supported by Japan Society for the Promotion of Science (JSPS) Grant-in-Aid for Scientific Research (KAKENHI) [grant numbers: JP16K10972(to TT), JP19K09381(to TT) and JP23K08360(to HS)]. The funders had no role in study design, data collection and analysis, decision to publish, or preparation of the manuscript.

**Competing interests:** The authors have declared that no competing interests exist.

showed that HSR could cause organ injury by activating inflammation [5–7]. Although there are several anti-inflammatory drugs in clinical use, effective pharmacological treatments for ALI remain elusive. Therefore, there is an urgent need to develop effective pharmacological therapeutic strategies to prevent HSR-ALI.

One of the drugs showing anti-inflammatory effects is carbamazepine (CBZ), although it is normally used as an anti-convulsant. Numerous studies have demonstrated the anti-inflammatory effects of CBZ in rats, specifically its ability to reduce inflammation caused by lipopolysaccharide (LPS) inoculation and to decrease subcutaneous carrageenan-induced inflammation [8–11]. However, no previous study has assessed the anti-inflammatory effects of CBZ on HSR-ALI. Additionally, in terms of its effect on organ injury, previous studies have reported that CBZ can ameliorate liver injury in rats [12–17]. In addition to its effect on the liver, a study indicated that CBZ could lessen bacterial load and stimulate adaptive immunity in a zebrafish model [18]. Another study reported that CBZ is effective for treatment and control of asthma, with minimal side effects [19]. Despite these findings, the impact of CBZ on HSR-ALI remains unexplored.

Autophagy, a major cellular process maintaining cellular homeostasis [20], has been linked to various lung diseases and shown to have anti-inflammatory effects [21–24]. A previous study reported that autophagy activation ameliorates lung injury and inflammation in sepsis [22]. Moreover, autophagy can limit the activation of inflammasomes [23]. However, few studies have explored the role of autophagy in HSR-ALI [25]. As CBZ is used as an autophagy activator [12, 13, 26], we aimed to determine whether CBZ has a protective effect on HSR-ALI by suppressing inflammation and activating autophagy under the hypothesis that pretreatment with CBZ would ameliorate HSR-ALI by suppressing inflammation and activating autophagy.

## Materials and methods

### Laboratory animals and drug preparation

The Animal Use and Protection Committee of the Okayama University School of Medicine, Japan approved this animal experiment (OKU-2021787, November 17, 2021). This study adhered to the Guidelines for the Care and Use of Laboratory Animals, as outlined by the ARRIVE guidelines [27] and 2020 AVMA euthanasia guidelines [28]. Male Sprague-Dawley (SD) rats (Clea Japan Inc, Tokyo, Japan) weighing 350–400 g were housed under a controlled temperature of 25°C. A total of 46 rats were kept in the breeding room. The lights were turned on/off to simulate a 12h/12h day/night cycle, during which the rats were given adequate food and water until the start of the experiment.

CBZ was stored at −20°C, with dimethyl sulfoxide (100% DMSO) as the solvent, to achieve a 25 mg/mL concentration. Specifically, 25 mg of CBZ was combined with 1 mL of DMSO in a 2.0 mL screw-cap tube and vigorously vortexed until homogeneous. The aliquots were stored at −20°C and used within one month. Before application, the product should be equilibrated to room temperature for at least one hour. For experimental administration, the stock solution was diluted with saline to a final concentration of 2.5 mg/mL CBZ. The final solution was warmed in a 50°C incubator to enhance solubility and ensure even distribution. The dosage for administration was 12.5 mg/kg body weight intraperitoneally. The solutions were prepared and used within one day to avoid degradation and maintain efficacy. A 10% DMSO solution was injected at a dose of 0.5 mL/kg body weight. Chloroquine (CQ) diphosphate salt (Sigma-Aldrich Co., St. Louis, MO, USA) was dissolved in saline to obtain a 2 mg/mL stock solution. The CQ dosage administered was 10 mg/kg body weight intraperitoneally. To ensure sufficient pharmacological action, the drug was administered intraperitoneally one hour before initiating the HSR model in the experimental rats.

## Experimental design and HSR protocol

The SD rats were randomly divided into control (SHAM) [n = 4–5 at each of the two different timepoints, total = 9], saline + HSR (HSR) [n = 3–6 at each of the two different timepoints, total = 11], CBZ + HSR (CBZ/HSR) [n = 5 at each of the two different time points, total = 10], DMSO + HSR (DMSO/HSR) [n = 5 at one-time point, total = 5], and CBZ + CQ + HSR (CBZ/CQ/HSR) groups [n = 3–6 at each of the two different timepoints, total = 11]. To determine whether post-shock administration has therapeutic effects, we administered CBZ to rats before resuscitation following shock and conducted pathological analyses (group HSR/CBZ, n = 5). The anesthetic drugs [0.3 mL medetomidine (1.0 mg/mL), 0.8 mL midazolam (5 mg/mL), and 1.0 mL butorphanol (5 mg/mL)] were added to 2.9 mL saline and subcutaneously injected into the rats at a dose of 0.2 mL/100 g body weight to maintain the rats in a drug-sleep state throughout the surgery. The rats were placed on a heating pad, the left femoral artery and vein were carefully dissected under sterile conditions, and polyethylene catheters were then inserted into both the artery and vein. After measuring the baseline blood pressure, blood was slowly drawn from the left femoral vein using a 1 mL sterile syringe until a mean arterial blood pressure of 30 ± 5 mmHg was achieved, avoiding a rapid drop in blood pressure that could cause death. During this period, the rectal temperature of the rat was continuously monitored. If the body temperature was too low, it was raised to the physiological level by turning on the heating pad, and if the body temperature was too high, it was lowered to the physiological level by general physical cooling. The body temperature of the rats was maintained within the physiological range. The shock lasted for 45 min. The rats were then resuscitated within 15 min by returning all the blood drawn until their blood pressure returned to baseline. The mean arterial blood pressure was maintained at baseline for 45 min. The sham-operated group (SHAM group) underwent the procedure manually, without blood sampling or transfusion. The animals were allowed to breathe independently throughout the experiment. After HSR, the animals were housed in a controlled temperature of 25˚C with adequate food and water. During this time, animals received no treatment. Following HSR, the rats were sacrificed at 3 h to check the expression of inflammatory factors or at 24 h to check the pathological changes and expression of autophagy-related proteins S2 File.

The rats were sorted according to their purchase order, marked, and housed in uniformly sized cages. The experiment was conducted at the same location, as described previously, using healthy and viable rats. Rats that died unexpectedly or exhibited signs of extreme weakness were excluded from the study. In addition, if any of the rats experienced intolerable pain during the experiment, we promptly discontinued the experiment and administered anesthesia with an overdose of isoflurane to minimize discomfort. Nevertheless, no rats fell into this category. All measures were implemented to ensure that the rats experienced minimal pain throughout the experiment.

## Histopathological analysis

The isolated lung tissues were fixed in 10% neutral buffered formalin solution for preservation. The tissues were removed for histological analysis and embedded in paraffin. Pathological sections of 5 μm thickness were prepared and clearly labeled by group. The sections were dewaxed, dehydrated, stained with hematoxylin and eosin (HE), and pathologically analyzed. Histopathological damage scoring and analysis were performed as previously described [5]. The lung histological changes were assessed by five blinded observers. Ten areas in each section were observed and scored according to the presence of vascular hemorrhage, tissue edema, inflammatory cell infiltration, and intra-alveolar hemorrhage: normal = 0, mild injury = 1, moderate injury = 2, and severe injury = 3 [29].

## Lung wet weight to dry weight (wet/dry) ratio

The rats were sacrificed via blood sampling from the abdominal aorta. The left lung tissue samples were blotted dry, wrapped in a dry and clean aluminum foil, weighed wet, and dried at 110˚C for 24 h. The wet/dry weight ratio was measured to quantify pulmonary edema.

## Lung tissue neutrophil count and myeloperoxidase (MPO) activity

Neutrophils in the lungs were stained with a naphthol AS-D chloroacetate staining kit (Sigma Diagnostics, St. Louis, MO, USA) using the lung tissue adjacent to that used in the histopathological analysis. Five non-consecutive sections of each rat were treated, and the number of neutrophils in the positively stained cells was observed at 200× magnification and counted by five blinded auxiliary investigators. For each experimental group, the data were averaged to obtain the final neutrophil count, representing the inflammatory response of each group. The enzymatic activity of MPO in the homogenized lung was measured by MPO assay kit (Cell Biolabs Inc., San Diego, CA).

## Reverse transcription-quantitative polymerase chain reaction (RT-qPCR)

Total RNA was extracted from the lung tissues. Total RNA was purified using the RNeasy® Mini Kit (Qiagen Sciences, Germantown, MD, USA). After removing potential contaminating DNA using DNase I (RNase-Free DNase set; Qiagen GmbH, Hilden, Germany), total RNA was reversed transcribed using a QuantiTect® Reverse Transcription Kit (Qiagen GmbH, Hilden, Germany) to generate first-strand cDNA. The PCR mixture was prepared using TB Green Premix Ex Taq II (Takara-Bio, Shiga, Japan). The PCR was performed using the Applied Biosystems StepOnePlus Real-Time PCR system (Thermo Fisher Scientific, CA, USA) as previously described [30]. The sequences of the upstream and downstream primers for tumor necrosis factor alpha (*TNF-α*), inducible nitric oxide synthase (*iNOS*), and *β-actin* were as follows: 5-GCCCTGGTATGAGCCCATGTA-3 and 5-CCTCACAGAGCAATGACTCCAAAG-3 for *TNF-α*; 5-CAAACTGTGTGCCTGGAGGTTC-3 and 5-AAGTAGGTGAGGGCTTGCCTGA-3 for *iNOS*; and 5-AACCCTAAGGCCAACCGTGAA-3 and 5-CAG GGACAACACAGCCTGGA-3 for *β-actin*. The mRNA levels of *TNF-α* and *iNOS* were normalized to the mRNA level of *β-actin*.

## Extraction of cytoplasmic proteins

The lung tissues were removed, cut, quickly frozen in liquid nitrogen, and stored at −80˚C. NE-PER nucleus and cytoplasm extraction reagent (Pierce, Rockford, IL, USA) was used to extract cytoplasmic and nuclear proteins according to the manufacturer's instructions. Pieces of the lung tissue stored at −80˚C were previously weighed, and those weighing approximately 100 mg were selected. The lung tissue was mixed with cytoplasm extraction reagent (CER)1 with protease inhibitor (cOmplete; Roche Diagnostics GmbH, Sigma-Aldrich, IN, Germany) at a dose of 1000 μL/100 g lung tissue weight, repeatedly milled into a homogenate, and then shaken thoroughly, followed by the addition of CER2 reagent at a dose of 55 μL/100 g lung tissue weight and thorough shaking. Thereafter, the mixture was centrifuged at $16,000 \times g$ for 5 min at 4˚C, and the supernatant was obtained as cytoplasmic protein fraction. Finally, the protein concentration (g/L) was determined using the BCA assay (Pierce, Rockford, IL, USA). The extracted cytoplasmic proteins were stored at −80˚C.

## Western blot analysis

Western blot analysis was performed after obtaining cytoplasmic proteins. After mixing, heating, and centrifugation, 20 mg/15 mL protein samples were applied to 10%, 12.5%, and 15%

(w/v) polyacrylamide-SDS gels for electrophoresis. Ten percent gels were used for P62, Beclin-1, and ATG12-ATG5; 12.5% for glyceraldehyde-3-phosphate dehydrogenase (GAPDH); and 15% for LC3 protein. After electrophoretic separation, the proteins were transferred onto an Amersham-Hybond polyvinylidene fluoride (PVDF) membrane (GE Healthcare Life Sciences, Germany). Protein blotting was performed after the completion of electrophoresis. After all proteins were transferred, the PVDF membrane was immersed in 4% (w/v) BlockAce solution at −4°C and closed overnight, followed by thorough rinsing with Tris-buffered saline with Tween 20 (TBS-T). After rinsing, the membranes were conjugated with primary and secondary antibodies.

GAPDH was used as an internal reference for comparison with target protein expression. Primary antibodies were diluted using TBS-T solution and added to PVDF membranes (LC3 primary antibody: rabbit anti-LC3 polyclonal antibody, 1:2000 dilution; MBL PM036, P62 primary antibody: rabbit anti-P62 polyclonal antibody, 1:1000 dilution; MBL PM045, GAPDH primary antibody: rabbit anti-GAPDH polyclonal antibody: sc-25778, SANTA CRUZ, 1:5000 dilution; mouse anti-ATG5 polyclonal antibody: sc-133158, Santa CRUZ, 1:1000 dilution; and mouse anti-Beclin1 polyclonal antibody: sc-48341, Santa CRUZ, 1:1000 dilution). The secondary antibodies (goat anti-mouse IgG-HRP: sc-2005; Santa Cruz, 1:10,000 dilution or goat anti-rabbit IgG-HRP: ab6721; Abcam, 1:10,000 dilution) were incubated at room temperature for 1 h. Thereafter, the membrane was rinsed again with TBS-T and placed in a detection reagent (Clarity Western ECL Substrate, Bio-Rad, USA) for 5 min. Finally, the antigen complexes were visualized using an image scanner (ChemiDoc XRS Plus Imaging System, Bio-Rad), and signal expression was quantified using an analysis software (Image Lab Version 5.0, Bio-Rad).

## Statistical analysis

Statistical analyses were conducted using GraphPad Prism version 9 (GraphPad Software, Inc., San Diego, CA, USA). Results were expressed as mean ± standard error (SEM). The Tukey-Kramer method was employed for multiple comparisons, and data were analyzed via ANOVA. A p-value < 0.05 was considered statistically significant.

## Results

### Effect of CBZ pretreatment on histological changes in the lungs of ALI rats

Microscopic examination (200× magnification) of the lung tissue sections stained with HE, 24 h post-resuscitation, revealed the impact of CBZ on HSR-induced ALI in rats. Histopathological analysis showed that the HSR group had more obvious tissue congestion, edema, inflammation, and hemorrhage than the SHAM group (Fig 1A). The lung tissue changes in the DMSO/HSR group were similar to those in the HSR group (Fig 1A). These histopathological changes markedly improved in the CBZ/HSR group (Fig 1A). However, the effects of CBZ were attenuated by the autophagy inhibitor, CQ (Fig 1A). In addition, the scores of pathological histological changes were significantly higher in the HSR group than in the SHAM group (*p* < 0.05), whereas the CBZ/HSR group had significantly lower lung injury scores than the HSR group (*p* < 0.05) (Fig 1B). The protective effect of CBZ was inhibited by CQ, resulting in an increase in the lung injury score (Fig 1B), suggesting that CBZ exerts a protective effect against HSR-ALI. Additionally, in the HSR/CBZ group, where CBZ was administered before resuscitation following shock, we observed significant improvements in tissue damage compared to the HSR group. This was evidenced by improvements in pathological changes and reduced lung injury scores. However, the extent of improvement was less pronounced compared to the group administered CBZ before HSR (S3 File).

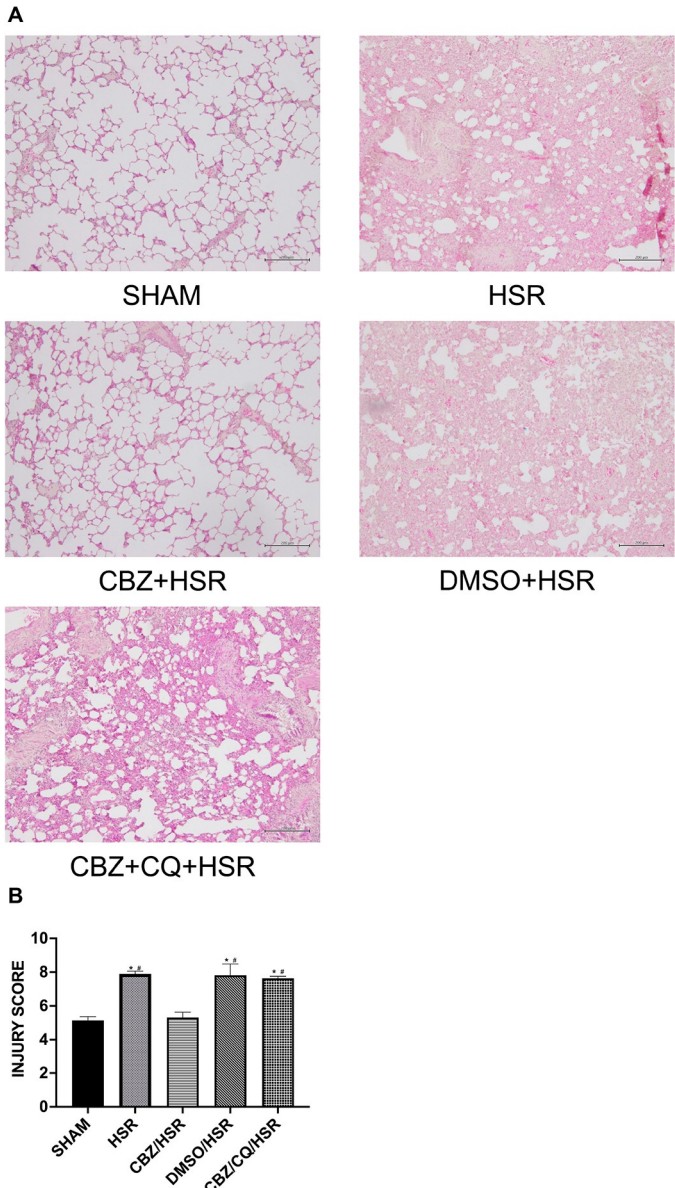

**Fig 1. Effect of CBZ pretreatment on HSR-induced histological damage in the lungs of rats with acute lung injury.**
Histological examination of lung injury in five groups. Rats were sacrificed 24 h after HSR, and lung tissues were collected for histological analysis. Lung sections were observed with the microscope (original magnification × 200). (A) Histological alterations including congestion, edema, inflammation, and hemorrhage were noted in the HSR group. However, these histopathological changes were significantly ameliorated after treatment with CBZ. DMSO had no influence on histological damage, while the CBZ/CQ/HSR group showed a severe damage compared with the CBZ/HSR group. (B) By analyzing the changes in pathological sections of each group, lung injury score was calculated. Consistent with histopathological results, lung injury scores were elevated in the HSR group; however, administration of CBZ markedly decreased these scores. DMSO had no influence on lung injury, while the CBZ/CQ/HSR group showed a higher injury score than the CBZ/HSR group. Data in each analysis are presented as the mean ± SEM (n = 5). *$p < 0.05$ vs SHAM, and #$p < 0.05$ vs CBZ/HSR. CBZ, carbamazepine; HSR, hemorrhagic shock and resuscitation; DMSO, dimethyl sulfoxide; CQ, chloroquine; HE, hematoxylin and eosin; SEM, standard error of the mean.

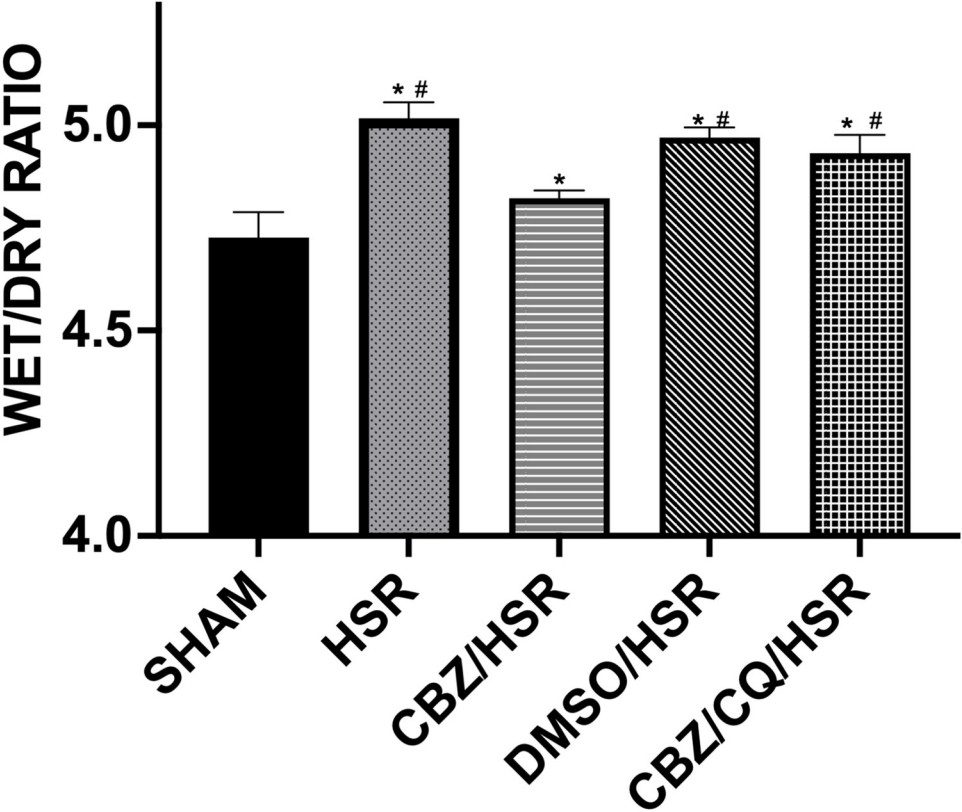

**Fig 2. Effect of CBZ pretreatment on pulmonary edema in rats.** Lung wet and dry weight ratio was calculated as an indicator of lung edema in each group. Rats were sacrificed 24 h after HSR, and lung tissues were collected. The HSR group showed a significantly higher lung wet and dry weight ratio; however, CBZ treatment significantly reduced the ratio. DMSO had no influence on the wet and dry weight ratio, while the CBZ/CQ/HSR group showed a higher wet and dry weight ratio than the CBZ/HSR group. Data in each analysis are presented as the mean ± SEM (n = 5). *$p < 0.05$ vs SHAM, and #$p < 0.05$ vs CBZ/HSR. CBZ, carbamazepine; HSR, hemorrhagic shock and resuscitation; DMSO, dimethyl sulfoxide; CQ, chloroquine; SEM, standard error of the mean.

### Effect of CBZ pretreatment on pulmonary edema in ALI rats

The analysis of lung wet/dry weight ratio, an indicator of pulmonary edema, showed the impact of CBZ on lung injury. The lungs in the HSR group had a significantly higher wet/dry weight ratio than those in the SHAM group ($p < 0.05$), and the DMSO/HSR group had a similar wet /dry weight ratio to the HSR group (Fig 2). Moreover, the CBZ/HSR group had a significantly lower wet/dry weight ratio than the HSR group ($p < 0.05$) (Fig 2). However, in the CBZ/CQ/HSR group, CQ abolished the positive effects of CBZ on lung tissue edema (Fig 2).

### Effect of CBZ pretreatment on neutrophil accumulation in the lungs of ALI rats

The analysis of neutrophil levels, an indicator of pulmonary inflammation, in the different groups showed the influence of CBZ on lung inflammation. Compared with the SHAM group, the HSR group showed a significant increase in the number of neutrophils and MPO activity ($p < 0.05$) (Fig 3). DMSO had no influence on neutrophil accumulation, similar to that in the HSR group (Fig 3). In contrast to that in the HSR group, lung neutrophil aggregation and MPO activity was significantly reduced in the CBZ/HSR group ($p < 0.05$) (Fig 3). The

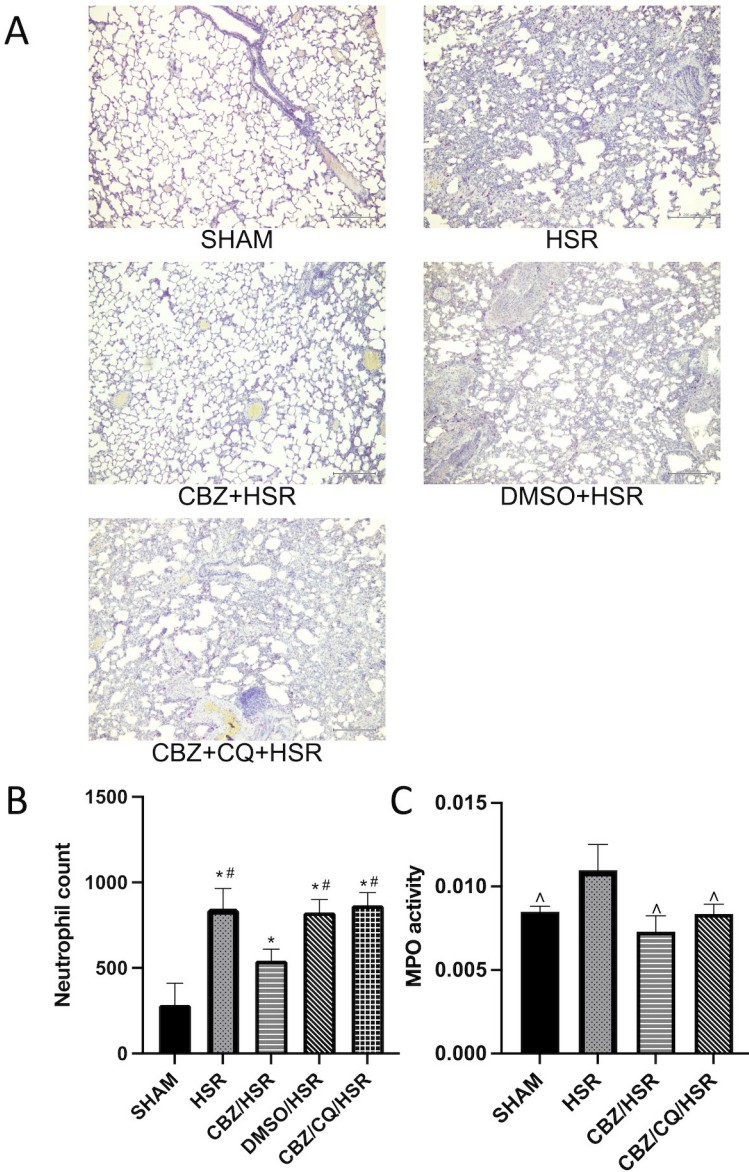

**Fig 3. Effect of CBZ pretreatment on neutrophil aggregation in the lung tissue of rats with acute lung injury.**
Neutrophil examination in each group. Rats were sacrificed 24 h after HSR, and lung tissues were collected for neutrophil staining and testing of MPO activity. Lung sections were observed with the microscope (original magnification × 200). (A) Neutrophil aggregation was obviously observed in the HSR group. However, the aggregation was significantly ameliorated after treatment with CBZ. DMSO had no influence on the neutrophil aggregation, while the CBZ/CQ/HSR group showed a higher neutrophil aggregation than the CBZ/HSR group. (B) Neutrophils were counted as described in the Materials and Methods sections. Consistent with the neutrophil aggregation, the HSR group showed a significantly higher number of neutrophils; however, CBZ treatment significantly reduced the number. DMSO had no influence on the neutrophil count, while the CBZ/CQ/HSR group showed a higher number of neutrophils than the CBZ/HSR group. (C) Enzymatic activity of myeloperoxidase in the lung homogenates. The HSR group showed a significantly higher level of myeloperoxidase, which was significantly reduced by CBZ treatment. Data in each analysis are presented as the mean ± SEM (n = 3~5). $^*p < 0.05$ vs SHAM, $^\#p < 0.05$ vs CBZ/HSR and $^\wedge p < 0.05$ vs HSR. CBZ, carbamazepine; HSR, hemorrhagic shock and resuscitation; DMSO, dimethyl sulfoxide; CQ, chloroquine; SEM, standard error of the mean.

autophagy inhibitor, CQ, increased the proliferation and aggregation of inflammatory cells (Fig 3), indicating that CBZ ameliorated lung injury by suppressing the inflammatory cells.

## Effect of CBZ pretreatment on the gene expression of the inflammatory mediators TNF-α and iNOS in the lungs of ALI rats

The analysis of gene expression levels of inflammatory mediators, including *TNF-α* and *iNOS*, indicators of pulmonary inflammation, in the different groups revealed the impact of CBZ on lung inflammation. The HSR group had significantly higher levels of *TNF-α* and *iNOS* expression than the SHAM group ($p < 0.05$) (Fig 4). CBZ treatment markedly downregulated the levels of *TNF-α* and *iNOS* mRNA ($p < 0.05$) (Fig 4). However, the autophagy inhibitor, CQ, abolished the effect of CBZ (Fig 4).

## Effect of CBZ pretreatment on autophagy in the lung tissue of ALI rats

Western blot analysis for autophagy marker proteins, including LC3, P62, Beclin-1, and Atg12-Atg5 conjugate, revealed the effect of CBZ pretreatment on autophagy in the lung tissue. LC3II/LC3I, Beclin-1, and Atg12-Atg5 levels were significantly higher in the CBZ/HSR group than in the HSR group ($p < 0.05$) (Fig 5). The protein expression level of P62, which is negatively related to autophagy, was significantly reduced in the CBZ/HSR group compared to that in the HSR group ($p < 0.05$) (Fig 5). In the experimental group treated with the autophagy inhibitor CQ, LC3II/LC3I and Beclin-1 were significantly decreased, while P62 was increased compared to that in the CBZ/HSR group, and Atg12-Atg5 showed a trend of decreasing, indicating that CBZ treatment promoted autophagy activation (Fig 5).

## Discussion

In this study, the comparison of pathological damage in the lung tissue and pulmonary edema between the groups showed that the administration of CBZ before HSR effectively protected rats against HSR-induced ALI; however, CQ pretreatment eliminated the protective effects of CBZ. Moreover, analysis of inflammatory mediator expression, such as *TNF-α* and *iNOS*, and neutrophil staining showed that CBZ pretreatment ameliorated the inflammation of the lung tissue. Additionally, the increase in the levels of autophagy-related proteins LC3, Beclin1, and Atg12-Atg5 and decrease in the level of P62 protein expression showed that CBZ had an impact on autophagy. Thus, our study data show that CBZ may improve HSR-induced ALI by inflammation inhibition, at least in part, through autophagy regulation.

CBZ belongs to a class of drugs known as anticonvulsants. CBZ is primarily prescribed to treat various neurological and psychiatric disorders and can be beneficial in improving many other types of diseases [14, 26, 31–33]. Moreover, CBZ can suppress inflammatory responses [8, 10, 11], which are closely related to ALI. However, research on ALI is still limited. Although some previous studies have focused on the effect of CBZ on respiratory-related diseases, most have concentrated on chronic respiratory diseases [34, 35]. Considering the positive effects of drug repositioning, we explored the protective effects of CBZ against HSR-induced ALI. Our study is the first to explore the protective effect of CBZ on ALI using an HSR model.

Our study showed that CBZ could ameliorate HSR-ALI, consistent with previous studies that showed the protective effects of CBZ against organ injury [12, 14, 18, 35]. One study indicated that autophagy initiation by CBZ leads to the removal of aggresome bodies and has the potential to suppress inflammatory response, apoptosis, and development of emphysema induced by cigarette smoke [35]. Moreover, CBZ treatment reduced bacterial burden, improved lung pathology, and stimulated adaptive immunity, thus improving tuberculosis infection [18]. CBZ has a protective effect against mouse liver injury induced by ischemia/

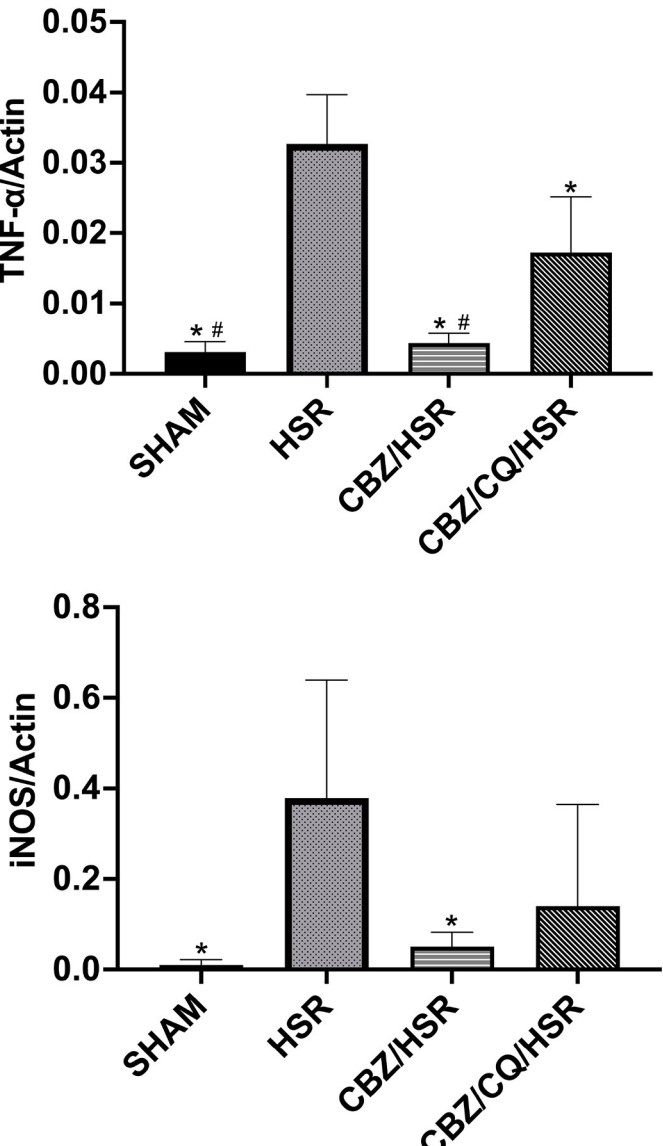

**Fig 4. Effect of CBZ pretreatment on gene expression of inflammatory mediators in the lungs.** The mRNA expression levels of TNF-α and iNOS by RT-qPCR analysis. Rats were sacrificed 3 h after HSR, and lung tissues were collected. The levels of TNF-α and iNOS were significantly higher in the HSR group. However, these changes were significantly reduced after treatment with CBZ. The CBZ/CQ/HSR group showed a higher level of TNF-α and iNOS than the CBZ/HSR group. Data in each analysis are presented as the mean ± SEM in arbitrary units (n = 4–6). *$p < 0.05$ vs HSR and #$p < 0.05$ vs CBZ/CQ/HSR.CBZ, carbamazepine; HSR, hemorrhagic shock and resuscitation; CQ, chloroquine; iNOS, inducible nitric oxide synthase; RT-qPCR, reverse transcription-quantitative polymerase chain reaction; SEM, standard error of the mean; TNF-α, tumor necrosis factor-α.

reperfusion via suppression of autophagy [12]. Consistently, CBZ exhibited a protective effect in our study. However, most previous studies used cell culture or clamped the lung hilum to simulate lung ischemia-reperfusion injury [36–40], which is different from our model. Our study indicated that CBZ improves HSR-induced ALI.

We investigated the mechanism of CBZ for improving lung injury and found that CBZ reduced the aggregation and number of neutrophils in the lung tissue. Moreover, CBZ down-regulated the expression of the genes *TNF-α* and *iNOS*, which are correlated with

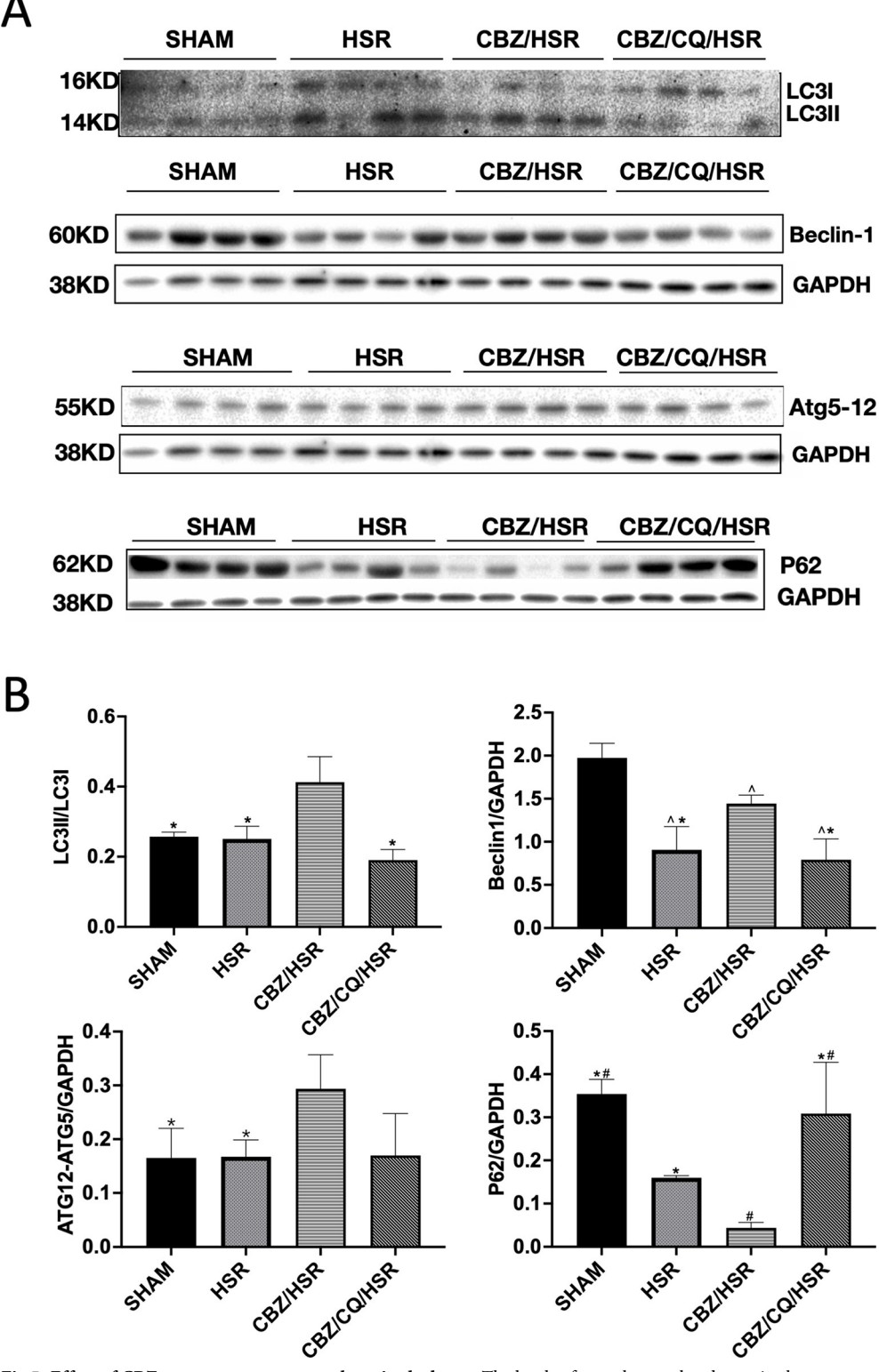

**Fig 5. Effect of CBZ pretreatment on autophagy in the lungs.** The levels of autophagy-related proteins by western blot analysis. Rats were sacrificed 24 h after HSR, and lung tissues were collected for western blot. The protein expression level was quantified by Image Lab Version 5.0 software. (A) Expression levels of the proteins LC3, Beclin1, P62, and ATG12-ATG5 conjugate were observed in each group. (B) Grayscale measurement of LC3II expression level normalized to LC3I expression level. Grayscale measurements of Beclin1, ATG12-ATG5, and P62 expression levels

were normalized to GAPDH expression level. Data are presented as mean ± SEM (n = 3–5). Statistical analysis was performed using ANOVA, followed by the Tukey–Kramer multiple comparison method between groups. $*p < 0.05$ vs CBZ/HSR, $^{\#}p < 0.05$ vs HSR and $^{\wedge}p < 0.05$ vs SHAM. CBZ, carbamazepine; HSR, hemorrhagic shock and resuscitation; CQ, chloroquine; SEM, standard error of the mean; GAPDH, glyceraldehyde-3-phosphate dehydrogenase.

inflammatory response. These results were consistent with those of previous studies. Bianchi found that CBZ exhibited a dose-dependent reduction effect on the inflammatory exudate, showing that CBZ possesses the capacity to impede the progression of various forms of inflammation in rats [10]. Moreover, CBZ could suppress the inflammatory mediators produced by stimulated glial cells [11]. Although these experiments used different models and applied animal and cytological experiments, the common result is that CBZ is able to inhibit inflammatory response [10]; consistently, our study showed that administration of CBZ attenuates the inflammatory response.

We investigated the effect of CBZ on autophagy by detecting the expression of autophagy-related proteins using an autophagy inhibitor. Autophagy plays an important role in lung injury [37, 41, 42]. Moreover, autophagy is a significant factor in the development of atopy and asthma, and its effects can vary, being either detrimental or advantageous depending on the specific cell types involved [43]. In addition, autophagy is related to chronic obstructive pulmonary disease (COPD), and one polymorphism of an autophagy gene (*Atg16L1*) is linked to a more than 3-fold increased risk of COPD [42]. However, these studies did not focus on the protective effects of autophagy-related drugs on diseases. As demonstrated in our study, CBZ consistently exhibited the ability to stimulate autophagy [15, 26, 44, 45]. Therefore, the protective effect of CBZ on ALI may be partially related to autophagy. Nevertheless, our investigation was an initial attempt to examine the impact of CBZ on autophagy in an HSR-ALI model. Multiple assays for autophagy exist, but in our study, we verified the changes in autophagy using common autophagy detection methods: detecting autophagy-related proteins and setting up an autophagy inhibitor group as a negative control [41, 46–48]. Although changes in autophagy have been evaluated, the role of autophagy in ALI requires further investigation.

Although important discoveries were made in our study, there are some limitations. Firstly, we pretreated with CBZ; however, the prediction of hemorrhagic shock beforehand in clinical practice is not easy. We tried post treatment of CBZ, but the therapeutic effect was not as significant as that of pretreatment. Therefore, different timepoints are required for CBZ treatment in the future to determine the optimal therapeutic window. Secondly, we employed a model of HSR-induced ALI having a milder degree of damage, which resulted in relatively low mortality. Therefore, there may be disparities compared with the clinical scenario. Thirdly, our diagnosis of ALI in rats differed from that in clinical settings. Because we did not monitor the respiration of rats or perform respiratory function tests, we were unable to diagnose ALI based on clinical diagnostic criteria, and we determined ALI by observing the appearance of the lung tissues after the experiments, staining the lung tissue slides, and detecting lung edema. Therefore, further studies are needed to incorporate respiratory function monitoring into our experimental model to make it more responsive to clinical situations. Fourthly, we detected autophagy using western blotting and an autophagy inhibitor that has been commonly used in previous studies. However, many other methods are available for autophagy detection. Future studies should incorporate additional methods to clarify the role of autophagy in ALI.

In conclusion, intraperitoneal pretreatment with CBZ significantly improved pulmonary injury and inflammation induced by HSR, at least partially, through enhanced autophagy. Thus, our study potentially offers a novel perspective for the treatment of ALI after HSR. Our

study will serve as a foundation for further studies on respiratory function monitoring in our model of HSR-induced ALI to make it more responsive to clinical situations and incorporating additional methods to clarify the role of autophagy in ALI. In future studies, we will investigate the effects of different administration times on therapeutic outcomes and examine the impact of CBZ on other organs in HSR rats, as this is an important research direction.

## Supporting information

**S1 File. Checklist.** This file contains The ARRIVE guidelines 2.0: author checklist.
(PDF)

**S2 File. Flow chart of the experiment.** This file depicts a timeline for the experimental procedure involving the administration of CBZ, CQ, and DMSO in this study.
(PDF)

**S3 File. Effect of CBZ treatment after shock on HSR-induced histological damage in the lungs of rats with acute lung injury (ALI).** Histological examination of lung injury in four groups. Rats were sacrificed 24 h after HSR, and lung tissues were collected for histological analysis. Lung sections were observed with the microscope (original magnification × 200). (A) Histological alterations, including congestion, edema, inflammation, and hemorrhage, were noted in the HSR group. However, these histopathological changes were significantly ameliorated after treatment with CBZ, while the administration of CBZ after shock also improved the tissue damage compared with the HSR group. (B) By analyzing the changes in pathological sections of each group, lung injury score was calculated. Consistent with the histopathological results, the lung injury scores were notably elevated in the HSR group; however, administration of CBZ before shock and after shock both markedly decreased these scores. Data in each analysis are presented as the mean ± SEM (n = 5). $^{*}p < 0.05$ vs SHAM, $^{#}p < 0.05$ vs HSR and $^{\nabla}p < 0.05$ vs CBZ/HSR. CBZ, carbamazepine; HSR, hemorrhagic shock and resuscitation; HE, hematoxylin and eosin; SEM, standard error of the mean.
(PDF)

**S4 File. Raw images.** This file shows the original uncropped and unadjusted images underlying all blot reports.
(PDF)

**S1 Table. Raw minimal data set.** This file contains supplementary data supporting the main findings of the study.
(XLSX)

## Acknowledgments

StepOnePlus used in this study belongs to the Central Research Laboratory, Okayama University Medical School.

## Author Contributions

**Conceptualization:** Toru Takahashi.

**Data curation:** Yaqiang Li.

**Formal analysis:** Yaqiang Li.

**Funding acquisition:** Hiroko Shimizu, Toru Takahashi.

**Investigation:** Yaqiang Li, Ryu Nakamura, Yifu Lu, Risa Sakamoto.

**Methodology:** Yaqiang Li, Hiroko Shimizu, Ryu Nakamura.

**Supervision:** Hiroko Shimizu, Ryu Nakamura, Emiko Omori, Toru Takahashi, Hiroshi Morimatsu.

**Writing – original draft:** Yaqiang Li.

**Writing – review & editing:** Hiroko Shimizu, Hiroshi Morimatsu.

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
