## [Decision Letter · Decision Letter 0]

26 Apr 2024

PONE-D-24-06929The protective effect of carbamazepine on acute lung injury induced by hemorrhagic shock and resuscitation in ratsPLOS ONE

Dear Dr. Shimizu,

Thank you for submitting your manuscript to PLOS ONE. After careful consideration, we feel that it has merit but does not fully meet PLOS ONE’s publication criteria as it currently stands. Therefore, we invite you to submit a revised version of the manuscript that addresses the points raised during the review process.

Both reviewers note that additional information is needed regarding drug dose and administration conditions. Additionally, Reviewer 1 raises concerns about the applicability of the present study. As an academic editor of PLoS One, I am aware that this is not a requirement for publication in this journal, and I think the discussion of limitations of your study in the Discussion section is adequate, but could be enhanced. You may also wish to consider the feasibility of conducting additional studies to enhance the study's impact and value. Reviewer 2 also provides suggestions for improving the overall readability of the manuscript.

We look forward to receiving your revised manuscript.

Kind regards,

Eric A. Shelden, Ph.D.

Academic Editor

PLOS ONE

Journal Requirements:

- https://doi.org/10.1038/s41598-023-31483-1

In your revision ensure you cite all your sources (including your own works), and quote or rephrase any duplicated text outside the methods section. Further consideration is dependent on these concerns being addressed.

3. To comply with PLOS ONE submissions requirements, in your Methods section, please provide additional information regarding the experiments involving animals and ensure you have included details on (1) methods of sacrifice, (2) methods of anesthesia and/or analgesia, and (3) efforts to alleviate suffering.

"This work was supported by Japan Society for the Promotion of Science (JSPS) Grant-in-Aid for Scientific Research (KAKENHI) [grant numbers: JP16K10972(to TT), JP19K09381(to TT) and JP23K08360(to HS)]."

6. We note that your Data Availability Statement is currently as follows: [All relevant data are within the manuscript and its Supporting Information files.]

7. PLOS ONE now requires that authors provide the original uncropped and unadjusted images underlying all blot or gel results reported in a submission’s figures or Supporting Information files. This policy and the journal’s other requirements for blot/gel reporting and figure preparation are described in detail at https://journals.plos.org/plosone/s/figures#loc-blot-and-gel-reporting-requirements and https://journals.plos.org/plosone/s/figures#loc-preparing-figures-from-image-files. When you submit your revised manuscript, please ensure that your figures adhere fully to these guidelines and provide the original underlying images for all blot or gel data reported in your submission. See the following link for instructions on providing the original image data: https://journals.plos.org/plosone/s/figures#loc-original-images-for-blots-and-gels. 

Reviewers' comments:

Reviewer's Responses to Questions

**Comments to the Author**

1. Is the manuscript technically sound, and do the data support the conclusions?

Reviewer #1: No

Reviewer #2: Yes

2. Has the statistical analysis been performed appropriately and rigorously? 

Reviewer #1: Yes

Reviewer #2: Yes

3. Have the authors made all data underlying the findings in their manuscript fully available?

Reviewer #1: Yes

Reviewer #2: Yes

4. Is the manuscript presented in an intelligible fashion and written in standard English?

Reviewer #1: Yes

Reviewer #2: Yes

5. Review Comments to the Author

Reviewer #1: [1] The manuscript by Li et al., describes the protective effect of CBZ on ALI induced by HSR in rats. While the data show protective effect in all the parameters used in this study, the question remains whether CBZ can be protective if given after the initiation of HSR and at the time of resuscitation. the manuscript will be significantly strengthened by examining the effect of CBZ at the time of resuscitation.

[2] In addition, details of the formulation of CBZ, concentration of DMSO used for the reconstitution etc are needed in the methods. It should also be clear in the methods that the drug was given prior to HS and resuscitation.

Lung MPO measurement is a good indicator for neutrophil infiltration in the lungs. I recommend the authors use either MPO in the lung tissues or Gr-1 histologically to assess neutrophil infiltration.

Reviewer #2: This seems like a well conducted and well written study. I have only minor comments:

Line 60: most studies referenced in 12-17 are mouse studies, so it’s not clear that the word “patients” is appropriate here.

Line 62: The paper cited used both zebrafish and mice. Zebrafish don’t have lungs, so I think this sentence needs to be revised.

Line 63: replace “another research” with “another study”?

Lines 88,89 – please provide the dosage of chloroquine used.

Line 353: “level of P62 protein expression”?

6. PLOS authors have the option to publish the peer review history of their article (what does this mean?). If published, this will include your full peer review and any attached files.

Reviewer #1: No

Reviewer #2: No

---

## [Author Response · Author response to Decision Letter 0]

13 Jul 2024

Reviewer #1: 

[1] The manuscript by Li et al., describes the protective effect of CBZ on ALI induced by HSR in rats. While the data show protective effect in all the parameters used in this study, the question remains whether CBZ can be protective if given after the initiation of HSR and at the time of resuscitation. the manuscript will be significantly strengthened by examining the effect of CBZ at the time of resuscitation. 

Thank you for the insightful comments and the constructive critique of our manuscript detailing the effects of carbamazepine (CBZ) on acute lung injury (ALI) induced by hemorrhagic shock and resuscitation (HSR) in rats.

Your suggestion to investigate whether CBZ could be protective if administered during the resuscitation phase raises an important consideration for our study. We acknowledge the potential significance of this timing on the therapeutic efficacy of CBZ and appreciate the depth it could add to our findings.

In response to your comments, we have conducted additional experiments. We administered CBZ at the beginning of resuscitation and compared the pathological changes. We found that CBZ treatment at the time of resuscitation could also improve lung injury, although pretreatment with CBZ showed a better therapeutic effect.

We hope this amendment strengthens the manuscript and paves the way for future research in this direction. Thank you once again for your valuable feedback, which has undeniably enriched our study.

[2] In addition, details of the formulation of CBZ, concentration of DMSO used for the reconstitution etc are needed in the methods. It should also be clear in the methods that the drug was given prior to HS and resuscitation.

Thank you for the detailed review and insightful feedback on the methodology section of our manuscript.

We acknowledge the need for clarity and detail regarding the formulation of carbamazepine (CBZ) and the concentration of DMSO used for reconstitution. In response to your comments, we have revised the methods section to include detailed information about the preparation of CBZ, specifying the concentration of DMSO used and the steps taken to ensure proper dissolution and stability of the solution.

Additionally, we have made it explicitly clear in the methods section that CBZ was administered to the experimental subjects prior to the initiation of hemorrhagic shock and resuscitation (HS). This clarification ensures that the timing of drug administration is transparent and accurately reflects the experimental design intended to evaluate the protective effects of CBZ.

We appreciate your guidance in enhancing the precision and reproducibility of our experimental procedures. We hope these modifications address your concerns and improve the methodological rigor of our study.

Thank you once again for your valuable contributions to the refinement of our work.

[3] Lung MPO measurement is a good indicator for neutrophil infiltration in the lungs. I recommend the authors use either MPO in the lung tissues or Gr-1 histologically to assess neutrophil infiltration. 

Thank you for the insightful comments and suggestions regarding our manuscript.

Regarding your recommendation to employ MPO or Gr-1 histological assessments for evaluating neutrophil infiltration in lung tissues, we recognize the value these additional experiments would add to our findings.

We checked MPO activity and compared it across different groups. Consistent with the neutrophil results, the CBZ treatment group showed a significant decrease in MPO activity compared to the HSR group.

Reviewer #2: 

This seems like a well conducted and well written study. I have only minor comments:

[1] Line 60: most studies referenced in 12-17 are mouse studies, so it’s not clear that the word “patients” is appropriate here.

Thank you very much for your thoughtful and constructive comments regarding our manuscript. Your feedback has been invaluable to refining our study and strengthening the presentation of our findings. 

Regarding your comment on Line 60, you rightly pointed out that the term "patients" may not be appropriate since the referenced studies primarily involve mouse models. After careful consideration, we have deleted this term to avoid confusion and more accurately reflect the nature of the studies involved.

[2] Line 62: The paper cited used both zebrafish and mice. Zebrafish don’t have lungs, so I think this sentence needs to be revised.

Thank you for your detailed review and insightful comments on our manuscript. We appreciate your guidance and the opportunity to improve our paper.

In response to your observation on Line 62, we have re-evaluated the section discussing the findings from the paper that utilized both zebrafish and mice. You correctly noted that zebrafish do not possess lungs, which is a critical anatomical distinction that impacts the interpretation of the results. We have revised this sentence to ensure its accuracy.

[3] Line 63: replace “another research” with “another study”?

Thank you for your detailed review and insightful comments on our manuscript. Regarding your suggestion on Line 63, we have replaced "another research" with "another study" to ensure the terminology is consistent and precise within the academic context. We agree that this change improves the formal tone and readability of our manuscript. We appreciate your guidance on these terminological adjustments and have carefully implemented them. We hope that these revisions meet your expectations and improve the overall quality of our paper.

[4] Lines 88,89 – please provide the dosage of chloroquine used.

Thank you for your meticulous review and attention to detail regarding our manuscript. In response to your comments on Lines 88 and 89, we have included the specific dosage of chloroquine used in our experiments. We have specified that the dosage administered was [10 mg/kg], which is consistent with the dosages commonly used in similar studies. This addition has been added to ensure the reproducibility of our results and to ensure full transparency regarding our experimental methods. We value your assistance in enhancing the accuracy and thoroughness of our study, and we hope that this addition meets the standards of the journal.

[5] Line 353: “level of P62 protein expression”?

Thank you for your careful attention to the details in our manuscript. In response to your comment on Line 353 concerning the "level of P62 protein expression," we have carefully reviewed our manuscript to ensure that the language accurately reflects the scientific measurements conducted. 

We appreciate your guidance in enhancing the scientific rigor of our paper.

---

## [Decision Letter · Decision Letter 1]

29 Jul 2024

PONE-D-24-06929R1The protective effect of carbamazepine on acute lung injury induced by hemorrhagic shock and resuscitation in ratsPLOS ONE

Dear Dr. Shimizu,

Thank you for submitting your manuscript to PLOS ONE. After careful consideration, we feel that it has merit but does not fully meet PLOS ONE’s publication criteria as it currently stands. Therefore, we invite you to submit a revised version of the manuscript that addresses the points raised during the review process. Specifically, please ensure that your manuscript, and especially the text added during the most recent revision, has been carefully edited, and please address the concerns of reviewer one in your discussion section.

We look forward to receiving your revised manuscript.

Kind regards,

Eric A. Shelden, Ph.D.

Academic Editor

PLOS ONE

Journal Requirements:

Reviewers' comments:

Reviewer's Responses to Questions

**Comments to the Author**

1. If the authors have adequately addressed your comments raised in a previous round of review and you feel that this manuscript is now acceptable for publication, you may indicate that here to bypass the “Comments to the Author” section, enter your conflict of interest statement in the “Confidential to Editor” section, and submit your "Accept" recommendation.

Reviewer #1: All comments have been addressed

Reviewer #2: (No Response)

2. Is the manuscript technically sound, and do the data support the conclusions?

Reviewer #1: Yes

Reviewer #2: Yes

3. Has the statistical analysis been performed appropriately and rigorously? 

Reviewer #1: Yes

Reviewer #2: Yes

4. Have the authors made all data underlying the findings in their manuscript fully available?

Reviewer #1: Yes

Reviewer #2: Yes

5. Is the manuscript presented in an intelligible fashion and written in standard English?

Reviewer #1: Yes

Reviewer #2: No

6. Review Comments to the Author

Reviewer #1: It is important to indicate in the discussion that the post treatment did not improve the outcome as similar to pre-treatment and consider this finding as a limitation of the study.

Reviewer #2: Overall, I think the manuscript has been improved, so I recommend accepting with minor revision. However, the text added to address the lack of experimental detail appears to have been copied from a protocol, and needs grammatical revision. Additionally, there are a number of other places where I think the manuscript would benefit from edits in order to improve clarity. Some examples are below. I note that there are numerous online editing services that might be employed to improve the readability of the manuscript.

57: no previous study has assessed

60: “and stimulate”?

61: and control of asthma

65: anti-inflammatory effects

86: until homogeneous

87: used within one month – otherwise, how was stability assessed?

89: the stock solution was diluted

89: The final solution was warmed

90: and used within one day to avoid degradation.

126: or at 24 h

7. PLOS authors have the option to publish the peer review history of their article (what does this mean?). If published, this will include your full peer review and any attached files.

Reviewer #1: No

Reviewer #2: No

---

## [Author Response · Author response to Decision Letter 1]

10 Aug 2024

Dear Reviewers,

We would like to express our sincere gratitude to you for taking the time to review our manuscript. We appreciate your insightful comments and suggestions, which have greatly helped us improve the quality of our paper. Below, we have addressed each point raised by you.

Reviewer #1:

Comment: It is important to indicate in the discussion that the post treatment did not improve the outcome as similar to pre-treatment and consider this finding as a limitation of the study.

Response: Thank you for this important observation. We have revised the discussion section to clearly indicate that the post-treatment did not yield an improved outcome compared to the pre-treatment. We have also acknowledged this as a limitation of our study. The relevant changes can be found in lines 429–431.

Reviewer #2:

Comment: Overall, I think the manuscript has been improved, so I recommend accepting with minor revision. However, the text added to address the lack of experimental detail appears to have been copied from a protocol and needs grammatical revision. Additionally, there are a number of other places where I think the manuscript would benefit from edits in order to improve clarity. Some examples are below. I note that there are numerous online editing services that might be employed to improve the readability of the manuscript.

Response: We appreciate your positive feedback and constructive suggestions. We have carefully revised the manuscript to address the grammatical issues and improve clarity. 

Specifically, we have sought the assistance of an online editing service to further improve the readability and overall quality of the manuscript.

Once again, we are grateful for your valuable feedback and the opportunity to improve our work. We hope that the revisions meet your expectations and that our manuscript is now suitable for publication.

Best regards.

Hiroko Shimizu

---

## [Decision Letter · Decision Letter 2]

15 Aug 2024

The protective effect of carbamazepine on acute lung injury induced by hemorrhagic shock and resuscitation in rats

PONE-D-24-06929R2

Dear Dr. Shimizu,

We’re pleased to inform you that your manuscript has been judged scientifically suitable for publication and will be formally accepted for publication once it meets all outstanding technical requirements.

Kind regards,

Eric A. Shelden, Ph.D.

Academic Editor

PLOS ONE

Additional Editor Comments (optional):

Reviewers' comments:

Reviewer's Responses to Questions

**Comments to the Author**

1. If the authors have adequately addressed your comments raised in a previous round of review and you feel that this manuscript is now acceptable for publication, you may indicate that here to bypass the “Comments to the Author” section, enter your conflict of interest statement in the “Confidential to Editor” section, and submit your "Accept" recommendation.

Reviewer #1: All comments have been addressed

Reviewer #2: All comments have been addressed

2. Is the manuscript technically sound, and do the data support the conclusions?

Reviewer #1: (No Response)

Reviewer #2: Yes

3. Has the statistical analysis been performed appropriately and rigorously? 

Reviewer #1: (No Response)

Reviewer #2: Yes

4. Have the authors made all data underlying the findings in their manuscript fully available?

Reviewer #1: (No Response)

Reviewer #2: Yes

5. Is the manuscript presented in an intelligible fashion and written in standard English?

Reviewer #1: (No Response)

Reviewer #2: Yes

6. Review Comments to the Author

Reviewer #1: (No Response)

Reviewer #2: All concerns have been adequately addressed to my satisfaction. I have no further concerns at this time.

7. PLOS authors have the option to publish the peer review history of their article (what does this mean?). If published, this will include your full peer review and any attached files.

Reviewer #1: No

Reviewer #2: No

---

## [Editor Report · Acceptance letter]

20 Aug 2024

PONE-D-24-06929R2 

PLOS ONE

Dear Dr. Shimizu, 

I'm pleased to inform you that your manuscript has been deemed suitable for publication in PLOS ONE. Congratulations! Your manuscript is now being handed over to our production team.

Kind regards, 

on behalf of

Dr. Eric A. Shelden 

Academic Editor

PLOS ONE